# Transcriptome Profiling of Embryonic Retinal Pigment Epithelium Reprogramming

**DOI:** 10.3390/genes12060840

**Published:** 2021-05-29

**Authors:** Jared A. Tangeman, Agustín Luz-Madrigal, Sutharzan Sreeskandarajan, Erika Grajales-Esquivel, Lin Liu, Chun Liang, Panagiotis A. Tsonis, Katia Del Rio-Tsonis

**Affiliations:** 1Department of Biology and Center for Visual Sciences at Miami University, Miami University, Oxford, OH 45056, USA; tangemja@miamioh.edu (J.A.T.); luzmadrigal@wisc.edu (A.L.-M.); sreesks@miamioh.edu (S.S.); esquiveg@miamioh.edu (E.G.-E.); liul2@miamioh.edu (L.L.); liangc@miamioh.edu (C.L.); 2McPherson Eye Research Institute, University of Wisconsin-Madison, Madison, WI 53705, USA; 3Waisman Center, University of Wisconsin-Madison, Madison, WI 53705, USA; 4Center for Autoimmune Genomics and Etiology, Cincinnati Children’s Hospital Medical Center, Cincinnati, OH 45229, USA; 5Department of Computer Science and Software Engineering, Miami University, Oxford, OH 45056, USA; 6Department of Biology, University of Dayton and Center for Tissue Regeneration and Engineering at the University of Dayton (TREND), Dayton, OH 45469, USA; ptsonis1@udayton.edu

**Keywords:** retina, RPE, reprogramming, EMT, RNA-seq, LCM

## Abstract

The plasticity of human retinal pigment epithelium (RPE) has been observed during proliferative vitreoretinopathy, a defective repair process during which injured RPE gives rise to fibrosis. In contrast, following injury, the RPE of the embryonic chicken can be reprogrammed to regenerate neural retina in a fibroblast growth factor 2 (FGF2)-dependent manner. To better explore the mechanisms underlying embryonic RPE reprogramming, we used laser capture microdissection to isolate RNA from (1) intact RPE, (2) transiently reprogrammed RPE (t-rRPE) 6 h post-retinectomy, and (3) reprogrammed RPE (rRPE) 6 h post-retinectomy with FGF2 treatment. Using RNA-seq, we observed the acute repression of genes related to cell cycle progression in the injured t-rRPE, as well as up-regulation of genes associated with injury. In contrast, the rRPE was strongly enriched for mitogen-activated protein kinase (MAPK)-responsive genes and retina development factors, confirming that FGF2 and the downstream MAPK cascade are the main drivers of embryonic RPE reprogramming. Clustering and pathway enrichment analysis was used to create an integrated network of the core processes associated with RPE reprogramming, including key terms pertaining to injury response, migration, actin dynamics, and cell cycle progression. Finally, we employed gene set enrichment analysis to suggest a previously uncovered role for epithelial-mesenchymal transition (EMT) machinery in the initiation of embryonic chick RPE reprogramming. The EMT program is accompanied by extensive, coordinated regulation of extracellular matrix (ECM) associated factors, and these observations together suggest an early role for ECM and EMT-like dynamics during reprogramming. Our study provides for the first time an in-depth transcriptomic analysis of embryonic RPE reprogramming and will prove useful in guiding future efforts to understand proliferative disorders of the RPE and to promote retinal regeneration.

## 1. Introduction

The retinal pigment epithelium (RPE) is a monolayer of pigmented cells located between the neural retina and choriocapillaris and constitutes the outer blood-retinal barrier. As such, the RPE plays an important role in visual disorders and has received special attention for its regenerative potential following retinal cell loss [1,2,3,4,5]. In humans, RPE plasticity has been observed during pathological conditions such as proliferative vitreoretinopathy (PVR), a defective repair process that leads to visual impairment [6]. The pathophysiology of PVR is a multifactorial process in which RPE cells dedifferentiate, shed epithelial characteristics, and acquire a mesenchymal phenotype in a process known as epithelial-mesenchymal transition (EMT) [7]. PVR can be incited as a repair mechanism following acute trauma, such as retinal detachment, but instead of regenerating lost retina, the RPE attempts wound repair via scar formation, ultimately resulting in visual impairment [6].

Mammalian RPE can generate neuronal or retinal cells under certain conditions. Between embryonic days 12 and 13, rodent RPE can be reprogrammed in vitro to form neural retina when treated with fibroblast growth factor 2 (FGF2) [8,9]. Similarly, human fetal RPE cells are able to proliferate and generate RPE stem-like cells that can be differentiated into RPE cells, rod photoreceptors, and mesenchymal lineages [10]. Moreover, RPE from adult humans can be reprogrammed with small molecules in vitro to form neuron-like cells that synthesize dopamine [11]. Together, the observations that human RPE responds to injury by undergoing EMT, as well as the capacity for fetal RPE cells to differentiate to retinal neurons, suggest that mature mammalian RPE may retain some capacity for producing neural retina. However, whether there is a specific combination of factors and conditions that can efficiently reprogram mature human RPE to retinal neurons remains unknown. 

The limited plasticity observed in mammals stands in contrast to organisms such as urodele amphibians and the chick embryo (Gallus gallus), which have the ability to fully regenerate their retina after damage [4,12,13,14]. The embryonic chick can regenerate its retina between embryonic day (E) 4–4.5 (Hamburger and Hamilton (H & H) stages 23–25) by reprogramming RPE cells into retinal progenitors that are capable of generating all major retinal cell types [4,12,15,16,17]. RPE reprogramming in the chick has been characterized at the molecular level in 3 distinct phases [18]. Phase I of RPE reprogramming can be incited by surgical retina removal (retinectomy) alone and occurs within 6 h of injury. During this phase, RPE cells dedifferentiate and transiently reprogram by up-regulating pluripotency-associated factors and eye field transcription factors, while simultaneously repressing RPE identity genes. However, RPE reprogramming stalls in Phase I unless an exogenous reprogramming-inducing factor, such as FGF2, is introduced at the time of retinectomy. In the presence of FGF2, these transient changes in gene expression are augmented and sustained, committing the RPE to reprogram. Importantly, we have previously shown that the RPE remains non-proliferative during Phase I, and only in the presence of FGF2 is the RPE able to proceed to a proliferative stage in Phase II [18]. During this phase, the onset of proliferation can be observed by 24 h post-retinectomy (PR), and a new neuroepithelium containing a pool of retinal progenitor cells forms within 3 days PR. Finally, during Phase III, the differentiation of retinal cells takes place within 7 days PR [15,18].

A number of different factors and pathways have been implicated in chick RPE reprogramming, but the central mechanisms underlying this process remain unclear. For instance, FGF2 induces RPE reprogramming through activation of the mitogen-activated protein kinase (MAPK) pathway, leading to the induction of the eye field transcription factor PAX6 and suppression of the RPE marker MITF [15,19]. Moreover, the induction of ectopic SHH signaling in the RPE can inhibit reprogramming by protecting the RPE identity [15]. Inhibition of β-catenin/LEF/TCF transcriptional activity is sufficient to induce RPE reprogramming, even in the absence of FGF2, demonstrating that a decrease in canonical WNT signaling occurs during reprogramming [20,21]. RPE cells can be reprogrammed in vitro at later stages (HH Stage 28–31) by using combined treatment with FGF2 and TGFβ inhibitors, suggesting that signaling by TGFβ family members also exert an active role in promoting RPE differentiation [9,22]. More recently, we demonstrated that the epigenetic landscape of RPE is dynamic during reprogramming, including histone modifications (H3K27me3/H3K4me3) and changes in DNA methylation profile, and that overexpression of DNA demethylation enzyme *TET3* is sufficient to reprogram RPE [23]. Despite efforts to characterize the various pathways which govern RPE identity, there has not been a study to date characterizing the global changes in gene expression, which occur during embryonic RPE reprogramming.

The intimacy of the RPE with surrounding tissues such as extraocular mesenchyme and retina has impeded efforts to profile the in-situ transcriptome associated with RPE reprogramming. Compounding this problem, we previously demonstrated that retinectomy leads to rapid changes in the transcriptome and epigenetic profile of the injured RPE cells, underscoring the need for tissue isolation techniques that do not introduce injury signals that may confound the analysis [18,23]. In order to avoid the use of dissociation protocols that could mask crucial processes, such as those influenced by cell-ECM interactions, we performed Laser Capture Microdissection followed by bulk RNA-sequencing (LCM-RNAseq). LCM-RNAseq allows for the capture of a precise and pure population of RPE cells without the disruption of the local tissue microenvironment, preserving the native transcript profile. In this study, we employed LCM-RNAseq to analyze the global transcriptome associated with early (Phase I) RPE reprogramming. We investigated the intact RPE as well as the transiently reprogrammed RPE (t-rRPE), which represents RPE 6 h following retinectomy, and reprogrammed RPE (rRPE), which represents RPE 6 h following retinectomy, additionally treated with FGF2. We identified a number of genes in the t-rRPE which have been previously associated with retinal injury, as well as novel processes associated with transient reprogramming, such as the induction of neural development genes and the suppression of proliferation-associated transcripts. In the rRPE, we identified a number of FGF-responsive genes and transcription factors which drive RPE reprogramming, as well as uncover previously unknown roles for EMT and extracellular matrix (ECM) remodeling in facilitating the early transition from RPE to neural retina progenitor.

## 2. Materials and Methods

### 2.1. Chick Embryos and Surgical Procedures

White Leghorn chicken eggs from Michigan State University were incubated in a humidified rotating incubator at 38 °C. At embryonic day 4 (H&H Stage 24), retinectomies were performed as previously described [15], and FGF2-soaked beads were introduced to the eye cup of embryos in the rRPE condition. Embryos were collected at 6 h PR and processed for LCM collection. 

### 2.2. Laser Capture Microdissection (LCM)

LCM collection was performed as previously described [18,24]. Briefly, embryos were collected and infiltrated with a sucrose gradient at 4 °C followed by 2:1 25% sucrose: OCT compound (Sakura Finetek, Torrance, CA, USA) for 1 h and snap-frozen in dry ice and methylbutane. Cryosections (12 μm) were collected on PEN Membrane Frame Slides (ThermoFisher Scientific, Waltham, MA, USA) followed by fixation in 70% ethanol at −20 °C for 30 sec. Slides were rehydrated in 4 °C DEPC-treated water for 30 s, stained with he-matoxylin (Millipore Sigma, Burlington, MA, USA) for 10 s, and dehydrated in a 70%, 95%, and 100% ethanol gradient for 30 s each. LCM was performed using a Veritas Laser Capture Microdissection system and software to collect RNA samples for library prepara-tion as described previously [18]. RPE was isolated from sections using IR capture and UV dissection on CapSure HS LCM Caps (ThermoFisher Scientific, Waltham, MA, USA), and total RNA extraction was performed using PicoPure RNA Isolation Kit (Applied Biosys-tems, Waltham, MA, USA), including a treatment with DNase I (RNase-Free DNase Set, Qiagen, Hilden, Germany). The quality and quantity of RNA were determined using Ag-ilent RNA 6000 pico kit (Thermo Fisher, Waltham, MA, USA) on the Agilent 2100 Bioana-lyzer.

### 2.3. mRNA Enrichment and RNA Sequencing Library Preparation

Total RNA was processed for library construction by Cofactor Genomics (St. Louis, MO, USA) according to the following procedure. Briefly, total RNA was reverse-transcribed using an Oligo(dT) primer, and limited cDNA amplification was performed using the SMARTer Ultra Low Input RNA Kit for Sequencing—v4 (Takara Bio, Shiga, Japan). The resulting full-length cDNA was fragmented and tagged, followed by limited PCR enrichment to generate the final cDNA sequencing library (Nextera® XT DNA Library Prep, Illumina, San Diego, CA, USA). Libraries were sequenced as single-end 75 base pair reads on the Illumina NextSeq500 per the manufacturer’s instructions.

### 2.4. Mapping of RNA-Seq Reads to the Chicken Genome

A total of 44–68 million single-end 75 bp reads were processed for each of 9 RNA-seq samples. Read quality was assessed using FastQC v0.11.8 and MultiQC v1.8 [25,26]. Reads were quality trimmed and adapter sequences removed using Trim Galore v0.6.1 and Cutadapt v1.18 with the parameters: --stringency 3 --length 36 [27,28]. Trimmed reads were aligned to *Gallus gallus* genome assembly GRCg6a using STAR v2.7.3, and splice sites were incorporated from Ensembl annotation 98 [29,30]. Aligned reads were sorted with Samtools v1.9 [31]. Quantification of transcripts was performed with Stringtie v2.0.4 using parameters and -A -C -B -e for unstranded libraries, and counts were summarized at the gene level [32].

### 2.5. Data Filtration and Differential Expression Testing

The resulting gene count matrix from Stringtie was passed to DESeq2 v1.22.2 for differential expression testing [33]. Low coverage genes were discarded from analysis (combined counts <20), and apeglm shrinkage of log2 fold change estimates were applied during differential expression testing [34]. Differentially expressed genes (DEGs) were defined as |log2(fold change)| ≥1 and adjusted *p*-value ≤ 0.05. All expression values are displayed in DESeq2 normalized counts or transcripts per million (TPM), as specified in the figures. The PCA analysis was performed on the DESeq2 normalized gene count matrix. Genome browser visualizations were rendered with the Integrative Genomics Viewer, using bam files normalized to TPM with the bamCoverage function of deepTools2.0 [35,36].

### 2.6. Clustering and Pathway Enrichment Analysis

Genes meeting the DEG criteria were clustered using affinity propagation clustering with the R package APCluster [37,38]. DEGs associated with pairwise comparisons or identified clusters were analyzed with g:Profiler for pathway enrichment analysis [39]. Enrichment results with adjusted *p* ≤ 0.05 and less than 2000 total term genes were included in the analysis. Gene Set Enrichment Analysis was performed using GSEA v4.0.3 [40,41]. EMT genes were obtained from dbEMT 2.0, a curated database of EMT-associated genes [42]. Chicken orthologs were identified using BioMart, yielding a final list of 839 EMT-associated genes. ECM-associated terms were obtained from the matrisome gene sets, and Gallus gallus Ensembl IDs were obtained for available orthologs using BioMart [43]. Figures were generated in the R environment, as well as using Cytoscape v3.7.2 [44,45].

## 3. Results

### 3.1. LCM-RNAseq of the Early Reprogrammed RPE

In order to analyze changes in the transcriptome during the early stages (Phase I) of RPE reprogramming, we performed LCM-RNAseq. For the analysis, we considered that the RPE was in close proximity with other tissues, such as the developing retina and extraocular mesenchyme. Additionally, previous studies have demonstrated the importance of the ECM during retina development and regeneration, as well as interactions between the RPE and Bruch’s membrane which influence RPE behavior and differentiation [3,6,46,47]. Therefore, to avoid disruptive tissue isolation protocols that may mask crucial processes, as well as to isolate a precise and pure population of RPE cells, we performed laser capture microdissection followed by RNA-seq (LCM-RNAseq) (Figure 1a) [48]. RPE cells from 12 µm-serial tissue sections (~*n* = 30) of 9 embryos (3 for each condition) were obtained from: (1) un-injured RPE at embryonic day 4 of development; (2) transiently reprogrammed RPE (t-rRPE) 6 h PR; and (3) reprogrammed RPE (rRPE) 6 h PR, with the addition of an FGF2-soaked bead immediately after retinectomy (Figure 1a,b). For rRPE, only RPE in the immediate proximity of an FGF2-soaked bead was targeted for laser capture. The 6 h PR time point encompassed Phase I of chick RPE reprogramming, representing an acute response to retinal injury during which t-rRPE and rRPE remain non-proliferative (BrdU-negative, p27Kip1-positive), as we have previously shown [18]. 

Total RNA extraction yielded ~10 ng of RNA for each sample and was used to generate libraries for low-input poly-A enrichment RNA sequencing. Bulk RNA-seq resulted in 44.5−67.5 million high-quality reads per sample with a genomic alignment rate of 91.5–94% (Appendix A). A three-dimensional representation of Principal Component Analysis (PCA) using normalized gene expression counts showed the samples forming three distinct clusters (Appendix A). In addition, unsupervised clustering of total normalized gene expression confirmed congruence among biological replicates (Appendix A). The genomic distribution of reads among the conditions was similar, with the majority of reads (64−69%) mapped to annotated genic regions of the *Gallus gallus* genome (Appendix A). An additional 22−26% of reads aligned to unannotated genomic regions, with another 3.52−3.95% of reads multi-mapping, 1.7−1.8% aligned to ambiguous features, and only 3.0−3.8% unaligned (Appendix A). Following TPM normalization, housekeeping genes GAPDH and SDHA showed uniform coverage and consistent expression across samples, reflecting unbiased amplification between libraries and suitable post-sequencing normalization measures (Appendix A). Consistent with previous results, we observed the down-regulation of RPE marker *MITF* in the t-rRPE and rRPE, as well as the up-regulation of retinal progenitor genes *ASCL1* and *VSX2* in the t-rRPE and rRPE, although *VSX2* was only modestly increased in the t-rRPE (log2[fold change] = 0.78, adjusted *p* = 0.08) (Figure 1c,d and Appendix A) [18]. Changes in expression of cell identity markers *MITF*, *ASCL1*, and *VSX2* in the t-rRPE, as well as the close spatial proximity of t-rRPE and rRPE in the PCA, lend further support for a model of RPE reprogramming which is partially initiated by retinectomy, even in the absence of exogenous FGF2 (Figure 1c,d and Appendix A).

We next turned our attention to differential expression testing in order to identify genes regulated during early RPE reprogramming. To conduct differential expression testing, we used the R package DESeq2 and defined differentially expressed genes (DEGs) by the cutoff criteria of |Log2(Fold Change) | ≥1 and adjusted *p*-value ≤0.05 [33,34]. To summarize each of the pairwise comparisons, we constructed MA plots displaying each gene’s log fold change plotted against average expression, revealing no readily apparent biases in log fold change estimates and highlighting the distribution of genes with an adjusted *p*-value ≤0.05 in red (Appendix A). Using this method, we identified in the t-rRPE and rRPE, 296 and 363 DEGs, relative to intact RPE, respectively; another 127 DEGs were identified between t-rRPE and rRPE. These DEGs were used for further analysis in the subsequent sections (Appendix A).

### 3.2. Injury-Responsive Genes Initiate Reprogramming and Suppress Proliferation in the t-rRPE

Of the 296 DEGs discovered in t-rRPE relative to intact RPE, we identified 175 up- and 121 down-regulated genes, which are summarized in a volcano plot with DEGs marked in red (Appendix A). A complete list of these injury-responsive DEGs is also provided (Appendix A). To visualize the top DEGs, we display heatmaps of the 25 down- and up-regulated genes, which exhibited the strongest evidence for differential expression, ranked by adjusted p-value (Figure 2a). Interestingly, many of these genes have been previously associated with eye development, injury response, and retinogenesis across broad contexts. We briefly discuss below notable genes, which warrant further consideration in connection to RPE reprogramming.

Among the top 25 up-regulated genes, we identified *STRA6*, which encodes a protein associated with retinol metabolism in the RPE, as well as lens-associated genes such as the crystallins *CRYAA* and *CRYAB*, the filensin *BFSP1*, and the avian D-crystallin *ASL1* (Figure 2a) [49,50,51,52,53,54]. Alpha crystallins *CRYAA* and *CRYAB* serve a dual role as heat shock proteins with anti-apoptotic effects and have been observed to be stress-responsive in human RPE in-vitro [55,56]. These findings may be reconciled with observations that RPE can form lentoid structures when maintained in culture conditions and may reflect a propensity for RPE to express genes associated with lens identity when placed in ectopic or otherwise stressful environments [57,58,59,60,61]. Similarly, we identified as a top up-regulated DEG *GADD45G*, which serves as a DNA repair factor that can promote cell cycle arrest under stressful conditions [62]. The dual-specificity phosphatase (DUSP) *CDC14A* which not only counteracts MAPK signaling but also plays a direct role in DNA repair and exit from mitosis, further displayed potent up-regulation [63,64,65]. Together, these results highlight a number of stress-responsive genes which may play determining roles in stalling cell cycle progression within the early t-rRPE, and potentially serve inhibitory roles to retina regeneration in the absence of FGF2.

In addition to the above injury-associated factors, we identified a number of up-regulated genes, which are involved in the formation of neural retina, lending further support to a model of RPE reprogramming, which is initiated by retinectomy alone. Among the top DEGs is *ASCL1* (Figure 2a and Appendix A), encoding a transcription factor necessary for Müller glia reprogramming in mice and zebrafish; it has previously been demonstrated that the overexpression of *ASCL1* alone can reprogram chick RPE cell cultures to retinal neurons [66,67,68,69,70,71,72]. Moreover, the transcription factor-encoding gene *STAT3* was also induced in the t-rRPE after injury and has been shown to play an important role in neuroprotection and retina regeneration in mice, zebrafish, and birds (Appendix A) [68,73,74,75,76,77,78,79]. It is possible that Müller glia and RPE reprogramming may share commonalities via ASCL1 or STAT3 regulatory mechanisms [75,79]. AXIN2, a negative regulator of canonical WNT/TCF, is also up-regulated, further emphasizing the importance of decreasing WNT signaling for the initiation of RPE reprogramming (Appendix A) [21,80,81,82,83,84]. Similarly, we identified the up-regulated gene Neuromedin-U (*NMU*), a neuropeptide, which is able to activate the extracellular signal-regulated kinase (ERK) pathway, as well as the transcription cofactors *BTG1* and *BTG2*, which are involved in neural precursor differentiation and the suppression of proliferation (Appendix A; Figure 2a) [85,86,87]. Together, these observations highlight key shifts in known modulators of RPE and retinal identity immediately following retinectomy.

In order to glean more comprehensive insight into the biological functions regulated in the t-rRPE, we performed pathway enrichment analysis on the identified DEGs (Figure 2b,c). As expected, we found terms associated with lens and eye development to be highly enriched among our up-regulated genes (including *BFSP1*, *ASL1*, *ASL2*, *CRYBA4*, *CRYAA*, *CRYBA2*, *CRYAB*, *NTRK3*, and *STRA6*) (Figure 2b). Interestingly, our up-regulated dataset was also associated with axon injury and regeneration, including the genes *KIAA0319*, *TNC*, *JUN*, and *NREP*, highlighting potential similarities between the injured RPE and neural injuries in other contexts (Figure 2b; Appendix A) [88,89,90,91]. Arginine biosynthetic process is also highly enriched and could likely reflect the role of ASL family proteins as crystallins during avian lens development. Intrigued by these findings, we further examined our gene list using the Enrichr tool and the Jensen DISEASES database, and found high combined enrichment scores of our DEG set with a number of eye pathologies, including retinal detachment (Appendix A) [92,93,94]. The extensive overlap between DEGs in the injured t-rRPE and genes associated with eye pathologies highlights the conserved and vital functions of many of the identified DEGs in maintaining eye physiology.

On the other hand, analysis of the 121 down-regulated DEGs bolsters the indication that cell proliferation is suppressed in the t-RPE. Several of the top down-regulated DEGs are important regulators of cell cycle progression, including *PCNA*, *E2F1*, *MCM2*, *CDK6*, and *RRM2* (Figure 2a). The broad suppression of cell cycle-related functions was further emphasized in the pathway enrichment analysis, which includes terms largely related to DNA replication and cell migration, as well as related terms such as cell cycle DNA replication, mitotic DNA replication, and pre-replicative complex assembly (Figure 2c, Appendix A). It is possible that the observed suppression of cell cycle-related transcripts plays a determining role in stalling the t-rRPE in a non-proliferative state and could prevent the t-rRPE from progressing past Phase I of reprogramming in the absence of FGF2. Interestingly, enrichment analysis suggested that genes associated with cell migration and motility were also down-regulated in the t-rRPE, including the semaphorin *SEMA3G*, as well as many structural components, including fibronectin *FN1*, collagen *COL1A1*, and cadherin *CDH5* (Figure 2a,c, Appendix A). Together, these results highlight a previously undescribed network of gene regulation that is incited by retinectomy, and reflects a dynamic regulation of RPE cell behavior that extends to proliferation, migration, and retinal identity.

### 3.3. Transcriptome Changes in the Early rRPE reflect MAPK Signaling Activity and AP-1 Transcription Factors

We next sought to evaluate transcriptional changes, which occur during early reprogramming in response to combined injury and FGF2 treatment. Using the same parameters described above, we identified 100 down- and 263 up-regulated genes in the rRPE relative to the intact RPE (Appendix A). To this end, we further displayed the top 25 down- and up-regulated genes, as ranked by adjusted p-value, in row-normalized heatmaps (Figure 3a). Interestingly, we observed up-regulation of both *FGFR1* and *FGFR4* in the rRPE (Figure 3a; Appendix A). We have previously shown that FGF2 acts through FGF receptors to stimulate the MAPK cascade, establishing that the FGF-FGFR-MEK-ERK pathway is necessary for the RPE reprogramming process [19]. In the same study, it was shown that blocking FGFR1/2 signaling during reprogramming leads to the reduced presence of PAX6, and also results in reduced proliferation of the rRPE at 24 h PR. The evidence presented here also implies that FGFR1 is regulated at the transcript level during reprogramming. Notably, we also identified *FGFR3* as a top down-regulated gene, further underscoring that FGF2 signaling entails the regulation of FGF receptor expression (Figure 3a).

Surprisingly, among our top up-regulated genes were four prominent members of the AP-1 transcription factor complex, *FOS*, *JUN*, *ATF3*, and *MAFA* (Figure 3a; Appendix A). The expression of AP-1 subunits has been shown to be MAPK-inducible, and together can regulate a broad range of cellular functions ranging from proliferation to differentiation in a context-dependent manner [95]. AP-1 transcription factor activity is influenced by the available protein subunits, binding site accessibility, and the present intracellular environment. Future studies may clarify a role for AP-1 transcription factors during RPE reprogramming. A number of other up-regulated genes were identified in the rRPE, which were not observed following retinectomy alone, such as tyrosine kinase signaling antagonists *SPRY1* and *SPRY4*, which control a wide range of biological functions such as metabolism and differentiation (Appendix A) [96]. Two DUSP family members were also up-regulated, *DUSP1* and *DUSP5*, which have the capacity to regulate the activity and strength of MAPK signaling (Appendix A) [97]. Interestingly, two members of the secreted ligands of TGF-β family were up-regulated, *BMP2* and *BMP4*, which is notable given the crucial role of BMP signaling in controlling the specification and identity of the RPE and neural retina (Appendix A) [84,98,99,100]. Importantly, *SMAD6*, an inhibitory signal transducer of BMP and TGF-β signaling, was also up-regulated (Appendix A) [101]. These observations suggest that the correct balance of BMP signaling is important for the specification of RPE and neural retina, and the mechanisms of action of BMPs might differ between development, injury response, and reprogramming contexts.

In agreement with our previous study, we observed increased expression of *GADD45Β* in the rRPE, and it is possible that *GADD45B* regulates DNA methylation to facilitate the epigenetic reprogramming of RPE (Appendix A) [23]. As noted prior, and in agreement with previous studies, the neural retinal progenitor marker *VSX2* was identified as one of the top up-regulated DEGs in rRPE (Figure 3a) [18]. Of note, interleukin 6 signal transducer (*IL6ST*) was also up-regulated, which is associated with the signal transduction of several cytokines, including IL6 (Appendix A). In this regard, we previously demonstrated that complement component C3a is sufficient to induce retina regeneration through activation of progenitor cells located in the ciliary margin via STAT3 activation, which in turn activates inflammation-responsive factors such as IL6 [76]. It is possible that IL6 acts in a paracrine mode to contribute to RPE reprogramming. Similar to the t-rRPE, *STAT3* remained up-regulated in the rRPE, and although STAT3 activity is largely regulated at the protein level, further studies are necessary to explore its function in this context (Appendix A).

As expected, pathway enrichment analysis identified a diverse set of overrepresented terms among the up-regulated genes in the rRPE, especially those related to phosphate metabolism and the MAPK cascade. Notable terms include regulation of MAPK cascade, regulation of the ERK1 and ERK2 cascade, and negative regulation of phosphate metabolic processes, which includes genes such as *SPRY4*, *PTPRO*, *ATF3*, *SOCS3*, *DUSP1*, *EZR*, *SPRY1*, *DUSP5*, and *PTN*. These observations further point toward activation of the MAPK cascade as a main driver of RPE cell proliferation and reprogramming. Interestingly, the rRPE was enriched for up-regulated genes associated with cell migration, cell motility, and cell population proliferation, indicating a trend toward the partial recovery of these processes with the addition of FGF2 (Figure 3b), as opposed to the t-rRPE, which displayed down-regulated genes associated with cell migration, motility, and proliferation (Figure 2c). Given this pattern, it is likely that the associated genes play indispensable roles in initiating a regenerative response in the injured RPE. The complete list of enriched pathways and their associated gene sets are included in Appendix A. 

In contrast to the up-regulated gene set, *MITF* was among the top 25 down-regulated genes in the rRPE, reflecting the loss of RPE identity that is a defining feature of reprogramming (Figure 1c and Figure 3a). Similar to the t-rRPE, the structural genes fibronectin *FN1* and *CDH5* remained down-regulated in the rRPE. A substantial number of transcription factors were among the top 25 down-regulated genes, including *ID2*, *HEY1*, *PITX1*, *ETS1*, and *BARX2*, providing further evidence of a changing regulatory landscape that accompanies early RPE reprogramming (Figure 3a). Pathway enrichment analysis of the down-regulated genes in the rRPE reflected processes associated with structural, metabolic, and transcriptional changes. A number of terms related to extracellular space were identified, as well as a significant enrichment of genes associated with primary metabolic processes. Another important term associated with cell structure, metalloendopeptidase inhibitor activity, was also enriched, including the down-regulated genes TIMP3 and RARRES1. Other down-regulated terms include regulation of nitrogen compound metabolic process and transcription by RNA polymerase II, which reflects changes in metabolism occurring during early embryonic RPE reprogramming (Figure 3c, Appendix A).

### 3.4. Gene Clusters Summarize the Transcriptional Dynamics of RPE Reprogramming and Associated Functions

We next sought to more precisely define the networks of gene expression that are regulated during RPE reprogramming, as we considered that a number of DEGs could demonstrate regulatory patterns that are not readily captured by pairwise analysis. To address this problem, we turned to affinity propagation clustering, a method of organizing DEGs into discrete groups based on similarities in expression pattern [37,38]. After compiling a list of all non-redundant DEGs across conditions (*n* = 592), we used affinity propagation clustering to organize the DEGs into five clusters, with each cluster resembling a distinct, archetypal pattern of gene expression (Figure 4a and Appendix A). In addition, we analyzed each of these clusters using pathway enrichment analysis and integrated the results into a unified network (Appendix A). These five clusters are particularly revealing in expounding upon the gene networks which differentiate the non-regenerative response of the t-rRPE from the rRPE’s regenerative outcome, and they lend insights to the mechanisms by which FGF2 signaling commits injured RPE to reprogram. 

Clusters 1 and 2 contain DEGs with expression patterns exclusively regulated in the rRPE, suggesting that the expression of these genes is directly or indirectly dependent on the presence of FGF2 (Figure 4a and Appendix A). Cluster 1 and 2 exhibit inverse patterns, with cluster 1 genes showing markedly elevated expression in the rRPE, and cluster 2 genes being distinctively down-regulated in the rRPE. Given these patterns, we anticipated these clusters to be of importance in effectuating processes downstream of FGF2 signaling, with genes in cluster 1 acting as determining factors associated with neural retina and genes in cluster 2 of importance in maintaining RPE identity. In line with these expectations, cluster 1 contains *SIX6*, an eye field transcription factor required for neural retina development, while cluster 2 contains the transcription factor *OTX2*, which is associated with early RPE identity and is acutely down-regulated in amphibian RPE during retina regeneration [102].

We previously demonstrated that MAPK signaling is the downstream effector of FGF2 signaling in rRPE, and accordingly, pathway enrichment analysis of cluster 1 revealed MAPK signaling to be a highly enriched term (Figure 4b; Appendix A) [19]. Cluster 1 encapsulates a number of MAPK signaling antagonists, such as *DUSP1*, *DUSP7*, *RASA2*, *SPRY1*, and *SPRY2*, as well as the phosphatase *PTPRO* (Figure 4a; Appendix A). The pathway enrichment term Tissue Development is also highly enriched in cluster 1, which included genes belonging to a number of prominent developmental signaling pathways, including *FGFR1*, as well as *TGFB3* and the downstream negative regulator *SMAD6*. Similarly, cluster 1 is also strongly enriched for genes associated with actin filaments and epithelial polarization, such as the cytoskeletal elements *ACTN1* and *EZR*, the myosin *MYO1E*, cytoskeleton regulator *LIMA1*, and actin-interacting proteins *FSCN1* and *SLC9A3R1* (Figure 4; Appendix A). When considering cluster 2, the role of FGF2 in inciting actin rearrangements is reinforced, which includes the actin-associated genes *CGNL1*, *RHOF*, and *FMN1* (Figure 4; Appendix A) [103,104,105]. Cluster 2 further suggest shifts in extracellular matrix composition and inflammation, containing collagen *COL8A1*, fibronectin *FN1*, fibrillin *FBN1*, interleukins *IL16* and *IL18*, as well as complement receptor *C1R* (Figure 4a; Appendix A). Despite these observations, pathway enrichment analysis of cluster 2 did not return any significantly overrepresented pathways at the stringent cut-off of adjusted *p* ≤0.05. However, given the known functions of these genes and their altered expression in the rRPE, it is likely they play significant roles in modulating injury response and priming the RPE for the formation of retinal progenitor cells.

The largest number of DEGs are found in cluster 3 (*n* = 202). These genes adhere to an expression pattern following modest up-regulation in the t-rRPE (i.e., in response to injury), and further up-regulation in the rRPE, suggesting that exogenous FGF2 treatment enhances, but is not solely responsible for, these genes’ expression. Similar to cluster 1, cluster 3 is highly enriched for MAPK-associated genes, demonstrating a significant enrichment of genes associated with the MAPK cascade and regulation of phosphorylation (Figure 4b). Genes associated with these processes included *SPRY4*, *ATF3*, *JUN*, *DUSP5*, *BMP2*, *FGFR4*, *STAT3*, and *MAP3K4* (Figure 4a; Appendix A), which are all directly a part of the MAPK signal transduction cascade or have been shown to be regulated by this pathway. Although we observe significant elevation of these genes’ expression in the t-rRPE and rRPE, many of these genes are also regulated at the protein level, and their activity is likely only partially explained by their transcriptional regulation. The observation that these genes are modestly elevated in the t-rRPE suggests that they may play a role in transient reprogramming or in priming the injured RPE toward neural retina production. This may additionally indicate that these genes are also regulated by some pathway other than FGF2 signaling, especially one that leads to a burst in MAPK activity in the t-rRPE. It is unclear if the mechanism regulating these genes is unrelated to FGF2, or if this could be caused by endogenous FGFs, albeit at a lower concentration, which was insufficient to induce committed reprogramming in the t-rRPE. Surprisingly, similar to cluster 1, we also identified enrichment of the term cell migration in cluster 3, including genes such as *CXCL14*, *CCL4*, *CCL19*, *VEGFD*, and *TGFB2*, further reinforcing that the RPE is primed for cell migration within the first 6 h of reprogramming (Figure 4b; Appendix A).

Genes in clusters 4 and 5 adhere to inverse expression patterns and encompass genes with markedly reduced expression in the t-rRPE (cluster 4), and conversely, genes with elevated expression in the t-rRPE (cluster 5). These patterns suggest that regulation of these gene clusters is unique to an injured transcriptome and can be reversed with the addition of FGF2, and thus are likely to be associated with the injured RPE state. In general, there was more variability in gene expression in clusters 4 and 5, and the relative enrichment was somewhat less pronounced than that observed in clusters 1–3 (Figure 4a and Appendix A). As expected, cluster 4 is overwhelmingly associated with genes involved with cell proliferation and DNA metabolism, including factors such as *MCM3*, *MCM4*, *MCM5*, *MCM10*, *ORC6*, *PCNA*, and *CDK6*, indicating that upon injury, the t-rRPE represses cell cycle-related factors, a trend that is partially reversed in the rRPE. In this regard, metabolic activity and cell cycle have been linked with reprogramming and differentiation, and it may be possible that a non-proliferative state may be a prerequisite to provide a window for RPE reprogramming [106,107,108]. Cluster 4 also contained a number of genes whose expression remains reduced in both the t-rRPE and rRPE, notably *HEY1*, *MITF*, and *PITX1*. On the other hand, cluster 5 contains a number of t-rRPE-enriched genes, which increase in expression with retinal injury alone, but which were also largely reduced in the rRPE. Terms associated with cluster 5 include structural constituents of eye lens, argininosuccinate lyase activity, and amino acid metabolism (Figure 4b, Appendix A). These terms can be explained by the elevation of lens-related crystallins, such as *CRYAA*, *CRYAB*, *CRYBA2*, *ASL1*, and *ASL2* in the t-rRPE. However, the function of crystallin genes in RPE is still yet to be determined, and it is still unclear if the crystallins are acting in accordance with their roles as heat shock proteins as a response to retinal injury. 

### 3.5. RPE Reprogramming Exhibits Qualities of an Epithelial-Mesenchymal Transition and Is Accompanied by Extensive Extracellular Matrix Remodeling

PVR occurs when human RPE responds to injury signals by undergoing EMT and forming fibrous epiretinal membranes. Known modulators of EMT are thought to play an inciting role in this pathology, as elevated TGFβ levels have been directly implicated in PVR incidence both in-vitro and in-vivo, and injuries to epithelial integrity are also thought to be an inciting factor [7,109,110,111,112,113,114]. PVR certainly highlights the cellular plasticity of RPE, but it is unclear why the mammalian healing response is limited to fibrotic wound closure. Interestingly, knockdown of the eye field transcription factor PAX6 in the newt RPE is sufficient to initiate a PVR-like fibrotic response, suggesting that there may also be a role for the regulation of EMT mechanisms in regenerative systems [3,115]. Given these observations, we were intrigued by our finding that *TGFB2* is up-regulated in the t-rRPE and rRPE, leading us to speculate a possible role for EMT in initiating embryonic RPE reprogramming (Appendix A). To test this hypothesis, we performed gene set enrichment analysis (GSEA) using dbEMT 2.0, a curated compendium of EMT-associated genes, from which we identified 839 chicken orthologs [40,42]. Surprisingly, GSEA revealed a significant enrichment of the EMT-associated genes expressed in the regenerative rRPE relative to the non-regenerative t-rRPE (Figure 5a,b). To better explain this observation, we performed GSEA on the remaining pairwise comparisons, which revealed an overall trend in which EMT-related genes were modestly depleted in the t-rRPE following injury, and conversely, were highly enriched in the rRPE in relation to RPE (Appendix A). Together, these results suggest a robust ability for the RPE to modulate the transcription of EMT-associated factors. This observation supports a model in which injury alone leads to the suppression of EMT transcripts, a trend, which is reversed in the presence of FGF2. This pattern was especially evident in the genes most dramatically repressed in the t-rRPE and elevated in the rRPE, such as *SNAI1*, *TGFBR2*, *ELK3*, *SMAD3*, and *TGFB3* (Figure 5b). It is possible that EMT is in part a shared mechanism for initiating both RPE fibrosis and regeneration, with differing context-dependent outcomes.

We next sought to identify changes in ECM structure that accompany the EMT response and may further reflect a change in cell identity. To accomplish this, we performed GSEA using the matrisome, a collection of gene sets encoding ECM components and regulators [43]. RPE behavior is heavily influenced by interactions with the basement membrane, and separation from the basement membrane is a hallmark of EMT [47,116]. GSEA revealed that genes associated with the basement membrane are enriched in the intact RPE relative to both t-rRPE and rRPE, suggesting that the initiation of RPE reprogramming entails the dissolution of the basement membrane following injury (Figure 5c and Appendix A). Notably, we observed a general depletion of basement membrane components during reprogramming, including collagens *COL6A3*, *COL4A1*, and *COL4A6*, laminins *LAMA2*, *LAMA4*, *LAMB2*, and *LAMC1*, netrins *NTN3* and *NTNG1*, and nidogens *NID1* and *NID2*.

Concomitantly, GSEA also revealed that within the rRPE there is a general shift toward the elevated expression of ECM remodeling components and secreted factors, emphasizing that reprogrammed RPE adopts an expression profile, which exerts an active role in reshaping ECM and the local tissue microenvironment. The matrisome category of ECM Regulators was significantly enriched in the rRPE (FDR = 0.006), which notably includes genes such as the matrix metalloproteinases *MMP9* and *MMP28*, lysyl oxidase-like genes *LOXL1* and *LOXL2*, the lysyl hydroxylase *PLOD1*, and metalloendopeptidase genes *ADAMTS9* and *ADAMTS15* (Figure 5d and Appendix A). Similarly, there was a high enrichment of secreted factors in the rRPE (FDR = 0.006), reflecting a group of genes that are likely to be broadly involved in reshaping the local ocular environment and potentiating cellular cues (Figure 5e). Genes in this category include *CXCL14*, *HGF*, *NGF*, *BMP2*, *WNT6*, *HBEGF*, and *TGFB3*, among others (Appendix A). Taken all together, these observations suggest that a major feature of RPE reprogramming includes the repression of genes associated with the RPE basement membrane, with a simultaneous shift toward the production of remodeling factors and secreted proteins. These results are consistent with a model in which RPE cells in Phase I begin to dedifferentiate, dampen expression of an epithelial gene program, prime themselves to dissociate from the basement membrane, and actively secrete factors that influence and reshape their immediate environment (Figure 6, Table 1). However, the extent to which the early mechanisms of RPE reprogramming overlap with EMT regulation during PVR are yet unclear, and it is also unknown how embryonic RPE ultimately averts fibrosis.

## 4. Discussion

The loss of retinal neurons is irreversible in adult mammals. Following retinal injury, the reparative contribution from mammalian RPE is limited to fibrotic wound closure, which can be readily observed as PVR-derived membranes formed following retinal detachment. An attractive approach to combating retinal cell loss would entail redirecting this fibrotic response toward a regenerative outcome, similar to the spontaneous regeneration of urodele amphibians or the FGF2-dependent processes during embryonic chick reprogramming. Here, we have for the first time conducted a study to uncover the global transcriptional changes associated with early embryonic RPE reprogramming, representing a crucial step in understanding the molecular events that make tissue reprogramming possible. The evidence presented here supports a model in which retinectomy alone initiates a partial reprogramming response, yet this response is not sufficiently robust to sustain committed reprogramming in the absence of exogenously supplied FGF2. 

There are potential therapeutic implications for a model in which RPE transiently reprograms following injury alone, and future studies on this topic will help elucidate the mechanisms which set apart the injury response observed during embryonic stages from the fibrotic outcomes following adult injury. In the RPE of spontaneously regenerating organisms, such as the newt, retinectomy alone is sufficient to induce a gene expression profile characteristic of retinal progenitors, including pluripotency factors and eye field transcription factors [117,118,119]. In previous studies, we demonstrated that this behavior partially holds true in the embryonic chick system but is also in part FGF2-dependent [18]. Here, we reaffirm many of these observations and extend the regulatory network to include other factors characteristic of neural injury and development. Interestingly, the current study further suggests an EMT program is repressed in t-rRPE and robustly activated in rRPE, including prominent regulators of EMT such as *TGFBR2*, *TGFB3*, and *SNAI1* [116]. Numerous studies have implicated TGFB signaling and EMT in the pathogenesis of PVR, and future studies which compare the EMT profile of embryonic and adult models of RPE injury will lend insight to mechanisms underlying the differing outcomes [110,112,113,114,120]. 

Accompanying the EMT program, we observed extensive regulation of ECM factors during embryonic reprogramming. The changes in ECM-associated gene expression can largely be characterized by repression of basement membrane-associated genes in reprogrammed conditions and an increased expression of remodeling factors in the rRPE. Extensive remodeling of the ECM is required during EMT to facilitate the separation of epithelial cells from the basement membrane. Interestingly, there is evidence to suggest a role for cell-ECM interactions in controlling the cell cycle reentry of amphibian RPE. For example, MMPs, which serve roles in the degradation of various ECM components, are up-regulated in regenerating Xenopus RPE and are required for RPE transdifferentiation in-vitro [121]. Given the importance of these interactions, we chose to employ LCM in this study, as it is highly desirable to protect the intact RPE from injury signals while still isolating the cell population with high precision. However, this technique is not without limitations and can suffer from reduced, degraded RNA yields, requiring relatively increased numbers of PCR cycles during library preparation, and ultimately may result in lowered sensitivity to DEG identification. Thus, the targets identified herein via LCM-RNAseq may serve as a non-exhaustive repertoire of genes regulated during early RPE reprogramming.

## 5. Conclusions

We present a comprehensive and in-depth analysis of the gene regulation which occurs during early embryonic RPE reprogramming, facilitated by the use of LCM to capture the native transcript profile. These findings shed light on the biological processes governing the transient, injury-induced reprogramming of the t-rRPE, and committed, FGF2-induced reprogramming of the rRPE. Gene-level analysis points toward extensive regulation of pathways and functions known to be essential to RPE-derived retinal regeneration, including cell proliferation, injury response, MAPK activity, and transcription factor dynamics. Importantly, we observe extensive regulation of EMT-associated genes and ECM remodeling factors in the t-rRPE and rRPE, suggesting that RPE-to-neural retina reprogramming may be facilitated by an EMT program that has been associated with fibrosis in other contexts (Figure 6, Table 1). In conclusion, the genes discussed in this study implicate a diverse set of processes with putative roles in governing embryonic RPE reprogramming and will serve as an important reference for future studies which may further explore the latent plasticity of the RPE.

## Figures and Tables

**Figure 1 genes-12-00840-f001:**
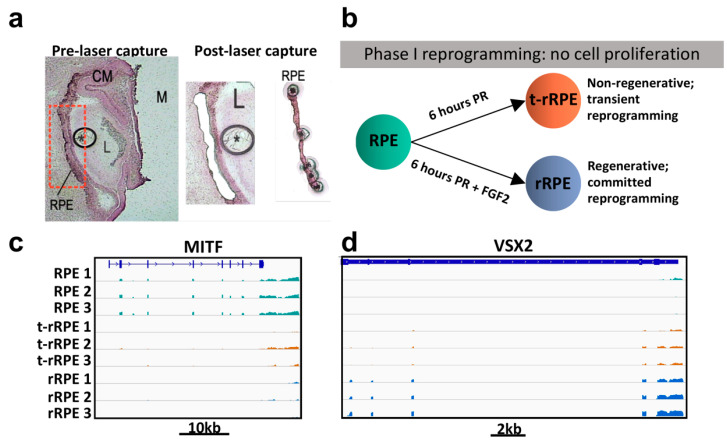
RNA-seq experimental design, quality control, and data are summarized. (**a**) RPE was isolated using LCM. (**b**) RNA-seq was performed on intact RPE, transiently reprogrammed RPE (t-rRPE) 6h post-retinectomy (PR), and reprogrammed RPE (rRPE) 6h PR in the presence of an FGF2-soaked bead. (**c,d**) Snapshot of Integrative Genomics Viewer genome browser of RPE-specific gene *MITF* and retinal progenitor gene *VSX2*, respectively. The scale bar below shows the genomic distance in base pairs. Labels: CM: Ciliary Margin; L: Lens; RPE: Retinal pigment epithelium; *: FGF2-soaked bead.

**Figure 2 genes-12-00840-f002:**
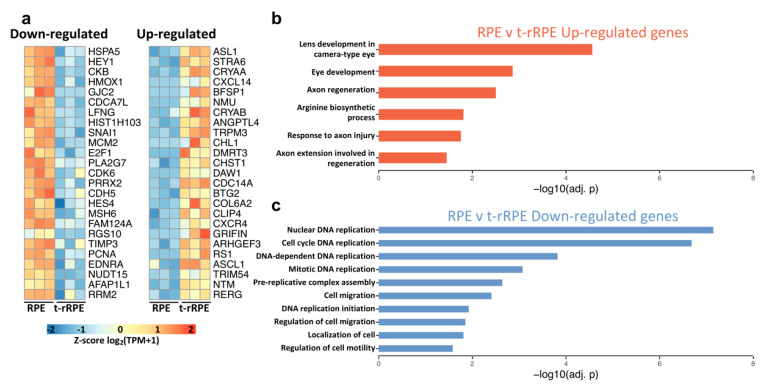
Injury-responsive DEGs control cell cycle dynamics and initiate reprogramming in the early t-rRPE. (**a**) Top 25 up- and down-regulated genes, as ranked by adjusted *p*-value, are displayed in heatmaps. Values are row-normalized. (**b**) Pathway enrichment analysis of up-regulated genes. (**c**) Down-regulated genes were used for pathway enrichment analysis, using the same criteria as in (**b**).

**Figure 3 genes-12-00840-f003:**
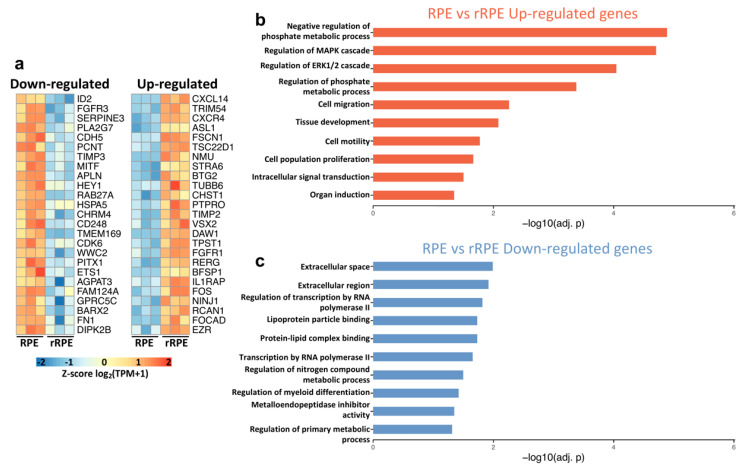
FGF2-responsive genes reveal a unique transcript profile in rRPE. (**a**) Top 25 up- and down-regulated genes are displayed in heatmaps, as ranked by adjusted p-value. Values are row-normalized. (**b**) Pathway enrichment analysis of the 263 up-regulated genes. (**c**) Summary of pathway enrichment results for the 100 down-regulated genes, created with the same criteria as in (**b**).

**Figure 4 genes-12-00840-f004:**
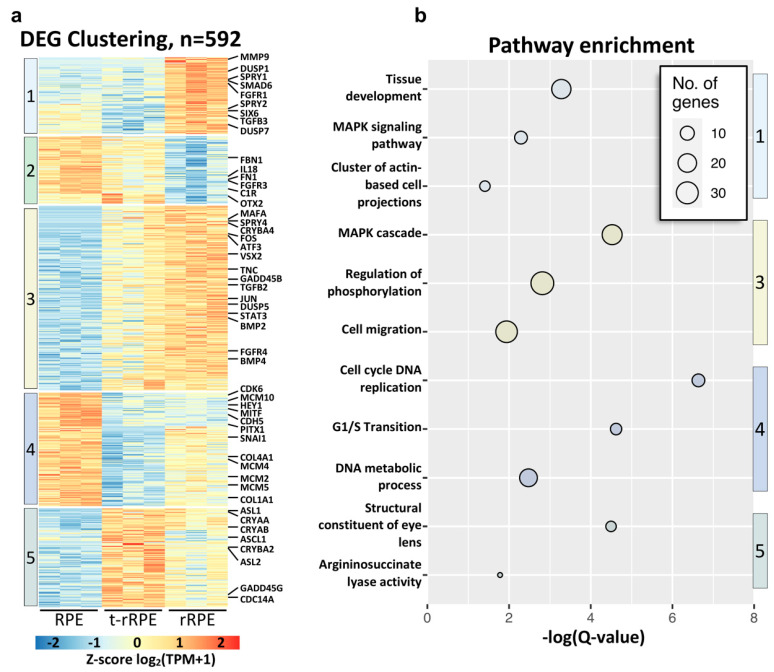
Clustering reveals gene expression dynamics during RPE reprogramming. (**a**) Affinity propagation clustering of 592 non-redundant DEGs delineated five gene clusters adhering to unique expression patterns. The heatmap is row-normalized and notable genes associated with retinal development and regeneration are annotated on the right. (**b**) Bubble chart displays key terms associated with each DEG cluster. The size of each bubble represents the number of DEGs identified for each term; the color of each bubble is associated with the cluster number from (**a**).

**Figure 5 genes-12-00840-f005:**
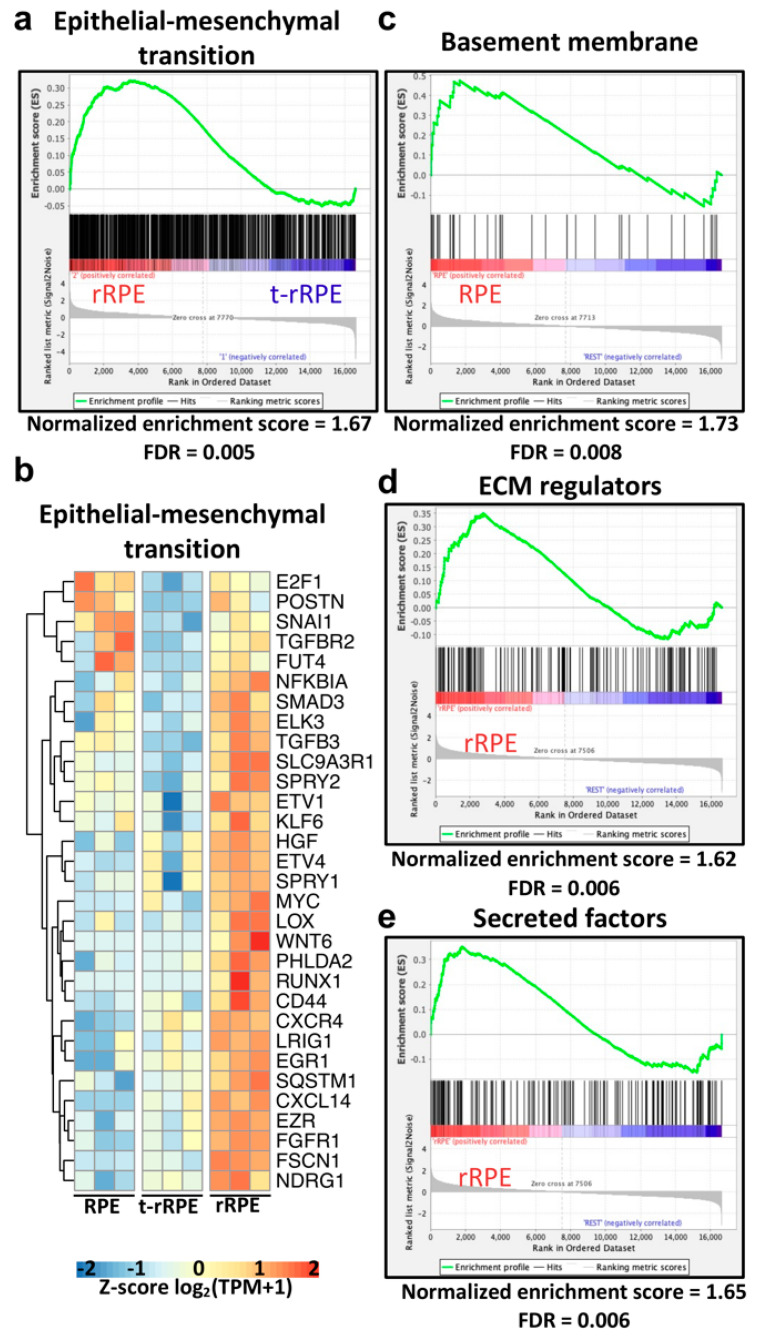
Epithelial-mesenchymal transition genes are highly enriched in the rRPE transcriptome, and extracellular matrix remodeling emerges as a defining feature of reprogramming. (**a**) Gene set enrichment analysis (GSEA) was conducted using the dbEMT 2.0 database and revealed significant enrichment (false discovery rate [FDR] = 0.005) of EMT-associated genes in the rRPE relative to t-rRPE. The GSEA report displays genes associated with the term as black bars, and the running enrichment score is shown above in green. (**b**) Leading edge genes identified in (**a**) are displayed in a row-normalized heatmap to visualize changes in expression. (**c**) GSEA using the Matrisome database revealed Basement membrane-associated genes were significantly enriched in the intact RPE relative to the other combined conditions (t-rRPE and rRPE). Conversely, the Matrisome database terms ECM regulators (**d**) and Secreted factors (**e**) were significantly enriched in rRPE relative to the other conditions (RPE and t-rRPE), suggesting that extensive remodeling of the extracellular matrix, including degradation of the basement membrane, occurs in the presence of FGF2.

**Figure 6 genes-12-00840-f006:**
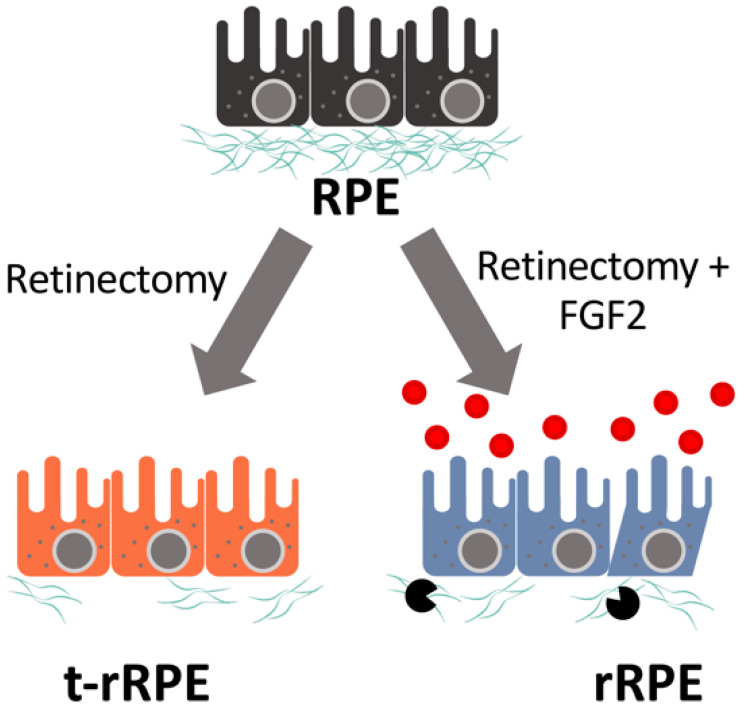
Schematic representation of the changes in gene expression. Major findings from the study are summarized in the diagram and table below, including the key biological processes associated with changes in gene expression in t-rRPE and rRPE. Example genes are listed for each category.

**Table 1 genes-12-00840-t001:** Associated functions in the t-rRPE and rRPE.

t-rRPE	rRPE
Reduced basement membrane synthesis *COL6A3, COL4A1, LAMA2*	Reduced basement membrane synthesis *COL6A3, COL4A1, LAMA2*
Reduction in EMT machinery *SNAI1, E2F1, TGFBR2, TGFB3*	Activation of EMT machinery *SNAI1, E2F1, TGFBR2, TGFB3*
Transient reprogramming *ASCL1*	Retinal progenitor profile *VSX2, ASCL1, SIX6*
Suppression of proliferation *MCM2/4/5/10, CDK6*	Production of secreted factors *PDGFD, CXCL14, WNT6, NGF, HGF*
Injury and damage-response *CRYAA, CRYAB, CRYBA2, GADD45G*	ECM remodeling *ADAMTS9, MMP9, LOXL2, PLOD1*
---	MAPK transcript profile *SPRY1/2, DUSP1/7*
---	AP-1 transcription factor expression *FOS, JUN, ATF3, MAFA*

## Data Availability

All code and resultant output files are available upon reasonable request to the authors. The data discussed in this publication have been deposited in NCBI’s Gene Expression Omnibus [122] and are accessible through GEO Series accession number GSE157129 (https://www.ncbi.nlm.nih.gov/geo/query/acc.cgi?acc=GSE157129, accesssed on 28 May 2021).

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
