# Peer review of "Transcriptome Profiling of Embryonic Retinal Pigment Epithelium Reprogramming"

_genes, 2021, doi:10.3390/genes12060840_

Round 1
Reviewer 1 Report
This manuscript contains probably important sets of data, however the authors failed to use these results. This reviewer completely agrees that it is important to compare gene profiling of three prepared condition to elucidate key molecules and pathways for neural regeneration. Obtained data could be informative, however the current manuscript needs to be improved.
1. Figures are too small to see, and submitted format of the manuscript looks wrong (Text should be presented in a right 2/3 page and figures/legend are in full-size page). Please consider how to present the figures, because the current figures are busy.
2. Description in the result section is not easily connected to the figures.
3. Table presentation of up-, down-regulated genes should be considered. Text with so many genes’ symbols makes the text complicated.
4. Conclusion is not supported by the presented data. The current conclusion sounds “discussion”.
5. Overall, this report is too hard to follow with many topics. This manuscript could be better by focusing more on the MAPK pathway, that the authors reported.
Reviewer 2 Report
In this manuscript Tangeman et al., used RNA-seq to characterize the transcriptome of embryonic RPE reprogramming. For that, authors used laser capture microdissection to isolate RNA from 1) intact RPE, 2) transiently reprogrammed RPE (t-rRPE) 6 hours post-retinectomy, and 3) reprogrammed RPE (rRPE) 6 hours post-retinectomy with FGF2 treatment. The study provides an in-depth transcriptomic analysis of embryonic RPE reprogramming and extensive description of the regulated genes and pathway involved.
One important aspect of this work is the fact that the transcriptome analysis was done in micro dissected tissue in intact RPE and after injury. This has generated a good resource of differentially expressed genes that set the basis for future projects to understand the contribution of these genes in reprogramming. The extensive bioinformatics analysis (that fills 5 figures) of relevant pathways (cell cycle, FGF genes..) in the reprogramming process will as well be useful for the scientific community in the field.
That being said, the work in this paper is descriptive, it does not provide validation of targets or provide novel mechanisms in reprogramming. However, the authors provide extensive description of the genes differentially expressed, and speculate about their potential implication in the reprogramming process, supported by published literature. Thus, I think this paper is a good source of information for the scientific community.
The printed figures are too small.
Reviewer 3 Report
The article 'Transcriptome profiling of embryonic retinal pigment epithelium reprogramming' is well written and structured. This will help the scientific community dealing with vision science mainly those who are involved in retinal pigment epithelium (RPE) research. The most important aspect of this study: it provides in-depth transcriptomic analysis of embryonic RPE reprogramming. It will be useful for the researchers to understand the proliferative disorders of the RPE and develop retinal regeneration methods.
The loss of retinal neurons is irreversible in an adult mammal. In this study, the global transcriptional changes associated with early embryonic RPE reprogramming could serve as a crucial step in understanding the molecular events that make tissue reprogramming possible. It may also unravel the mechanism for the scientific community that relates to the function and differentiation of the RPE.
Round 2
Reviewer 1 Report
Readability of the revised manuscript has been significantly improved.